# Use of New Glycerol-Based Dendrimers for Essential Oils Encapsulation: Optimization of Stirring Time and Rate Using a Plackett—Burman Design and a Surface Response Methodology

**DOI:** 10.3390/foods10020207

**Published:** 2021-01-20

**Authors:** Chloë Maes, Yves Brostaux, Sandrine Bouquillon, Marie-Laure Fauconnier

**Affiliations:** 1Institut de Chimie Moléculaire de Reims, UMR CNRS 7312, Université Reims-Champagne-Ardenne, UFR Sciences, BP 1039 boîte 44, CEDEX 2, 51687 Reims, France; sandrine.bouquillon@univ-reims.fr; 2Laboratoire de Chimie des Molécules Naturelles, Gembloux Agro-Bio Tech, Université de Liège, 2 Passage des Déportés, 5030 Gembloux, Belgium; marie-laure.fauconnier@uliege.be; 3Unité de Statistique, Informatique et Modélisation appliquées, Gembloux Agro-Bio Tech, Université de Liège, 2 Passage des Déportés, 5030 Gembloux, Belgium; y.brostaux@uliege.be

**Keywords:** essential oil, encapsulation, controlled release, biosourced, surface response methodology

## Abstract

Essential oils are used in an increasing number of applications including biopesticides. Their volatility minimizes the risk of residue but can also be a constraint if the release is rapid and uncontrolled. Solutions allowing the encapsulation of essential oils are therefore strongly researched. In this study, essential oils encapsulation was carried out within dendrimers to control their volatility. Indeed, a spontaneous complexation occurs in a solution of dendrimers with essential oils which maintains it longer. Six parameters (temperature, stirring rate, relative concentration, solvent volume, stirring time, and pH) of this reaction has been optimized by two steps: first a screening of the parameters that influence the encapsulation with a Plackett–Burmann design the most followed by an optimization of those ones by a surface response methodology. In this study, two essential oils with herbicide properties were used: the essential oils of *Cinnamomum zeylanicum* Blume and *Cymbopogon winterianus* Jowitt; and four biosourced dendrimers: glycerodendrimers derived from polypropylenimine and polyamidoamine, a glyceroclikdendrimer, and a glyceroladendrimer. Meta-analysis of all Plackett–Burman assays determined that rate and stirring time were effective on the retention rate thereby these parameters were used for the surface response methodology part. Each combination gives a different optimum depending on the structure of these molecules.

## 1. Introduction

For the last 70 years, industrial countries intensively used chemical pesticides in order to increase agricultural yields to feed a constantly growing population. Unfortunately, with time passing, controversies and the knowledge about their harmful effects on human health and environment have blown up quickly [1]. In this context, biopesticides are priceless candidates to create new weeds- and crops-managing strategies. Among natural compounds from plant origin, essential oils (EOs) are increasingly used for their various biological properties [2,3].

Essential oils are natural mixtures of volatile compounds frequently used in cosmetics, perfume, and sanitary products for both their fragrance and biological activities [4,5,6,7]. Another principal characteristic of EOs is their volatility, which limits residues after treatment. Unfortunately this can be a constraint for their utilization as biopesticide because their spread is not controlled [8]. To counter this, scientists developed several different encapsulation techniques. Depending on their properties, emulsion, coacervation, spray drying, complexation, ionic gelation, and nanoprecipitation help maintain a controlled release of EOs, either quick or slow [9]. EOs encapsulation may appear useless to enhance herbicidal activities on plants, because shoot death occurs after 1 h to 1 day of application [10]. However, an actual interest exists for the improvement of the seed’s germination inhibition effects because this one occurs for longer periods (up to 30 days) thus EO encapsulation with controlled release allows to use a lower concentration. Lethal dose depends on the target plant/seed [11].

Cinnamon and Java citronella essential oils are of particular interest for herbicidal applications in a context where the replacement of conventional herbicides is increasingly wanted [12,13,14]. In a previous study [12], we determined that the major constituents in cinnamon essential oil are trans-cinnamaldehyde (70%) followed by eugenol, caryophyllene, cinnamyl acetate, and linalool in decreasing concentration order. Java citronella EO is constituted of 57 different molecules; among them citronellal (40%), geraniol (20%), citronellol (15%), limonene (5%), and eugenol (2%) are the main representatives [12,15,16]. The modes of action of the main constituents of these EOs as herbicides are not fully characterized but their interaction with respectively the lipid and protein fraction of the plant plasma membrane might be involved [12].

In the present research, glycerol-based dendrimers (GDs) are proposed as new and original matrix to encapsulate EOs. GDs are macromolecules synthesized from glycerol carbonate (a side product from biofuel production) which already showed good encapsulation ability of contrast agent for medical sectors, metals (nanoparticles), and organic pollutants of used water. Indeed, their tree structure allows intern cavities (Figure 1), from various sizes depending of the dendrimer generation, to retain molecules [17,18,19,20]. Glyceroclikdendrimer (GAD) and glyceroladendrimer (GCD) have been recently developed and described in two patents with specific encapsulation abilities toward organic pollutants and metallic salts [21,22]. Beyond the agronomic field, EOs encapsulation within dendrimers can be used in a wide range of applications, including food industry (active packaging) and pharmaceutical (drug delivery system) through their bactericidal, viricidal, and fungicidal activities [23,24]. 

The goal of this study is to optimize the encapsulation reaction of two essential oils by four selected dendrimers by maximizing the retention of two GDs, a GCD and a GAD using a Plackett–Burman design (PBD) and response surface methodology (RSM) in order to eventually create an effective biosourced herbicide or for other applications where a slow release of EOs is required. PBDs are a screening design that takes into account a large number of factors with a minimal number of trials, while RSMs are an experimental design intended to optimize factors and their combinations [25]. Obviously, since this study highlights the statistical optimization of the encapsulation, these results can be applied in other fields cited before such as food preservatives creations [26].

## 2. Materials and Methods 

### 2.1. Chemicals and Reagents

The essential oils of *Cinnamomum zeylanicum* Blume bark (Cinnamon, CAN) and *Cymbopogon winterianus* Jowitt leaves (citronella, CIT) were purchased from Pranarom (Belgium).

Glycerodendrimers-polypropilenImine (GD-PPI) and glycerodendrimers-polyamidoamine (GD-PAMAM) were synthesized according the previously described work related to the decoration of dendrimers [17,18]. 

GlycerolADendrimers (GAD) and GlyceroClickDendrimers (GCD) were synthesized following the procedures described in two patents [21,22].

### 2.2. Essential Oils Encapsulation

Essential oils encapsulation take place by a spontaneous complexation; the dendrimers were dissolved in H_2_O (8 mL) and EOs were dissolved in ethanol (various concentrations). EOs solutions or pure ethanol was added to dendrimers solution (3/1 *v*/*v*) in a 22 mL glass vial which was directly hermetically sealed with a Teflon cap and covered with an aluminum foil to avoid light interference. Solutions were then stirred for at least 10 min at 100 rpm. According on the stirring settings, an emulsion of EOs occurs in the dendrimer solution, which provides a liquid phase EOs retention. This retention leads to a change in dynamic balance between solution and headspace compared to free EO solution (control), which is quantified by the following analysis. 

### 2.3. Dynamic-Headspace Gas Chromatography–Mass Spectrometry (DHS-GC–MS) Analysis

The percentage of retention (r) of EOs by GDs was determined by dynamic head- space sampling (DHS, Gerstel, Germany) coupled to a thermal desorption unit (TDU, Gerstel, Germany), a gas chromatograph (Agilent Technologies 7890A), and a mass spectrometer (MS, Agilent Technologies 5975C). During treatment in the DHS unit, the vials were conditioned at 25 °C for 30 min with stirring (500 rpm). The head-space sampling was performed on Gerstel TDU desorption tubes (OD 6.00 mm, filled with 60 mg of Tenax TA, Gerstel, Germany), on 200 mL at 20 mL/min, followed by 200 mL at 60 mL/min of drying phase. Desorption then occurred for 10 min at 300 °C and coupled to a cooled injection system (CIS, Gerstel, Germany) set at −80 °C. EOs were then transferred to the GC column (VF-WAXms, Agilent technologies USA; 30 m length, 0.250 mm I.D, 0.25 l m film thickness) for separation with temperature program as follow: Java citronella—from 70 °C (5 min) to 100 °C at a rate of 8 °C/min, then 2 °C/min to 160 °C, and then 20 °C/min to 260 °C (10 min); Cinnamon—from 40 °C (4 min) to 80 °C at a rate 3.5 °C/min, then 5 °C/min to 160 °C, and then 20 °C/min to 220 °C (10 min) with helium as carrier gas at a flow rate of 1.5 mL/min. The MS were recorded in electron ionization mode at 70 eV (scanned mass range: 35 to 300 m/z); source and quadrupole temperature at 230 °C and 150 °C respectively. The component identification was performed by comparison of the recorded spectra with two data libraries (Pal 600K^®^ and Wiley275) and injection of pure commercial standards in the same chromatographic conditions.

The percentage of retention (r) of EOs by GDs was calculated by the equation [27]:(1)$$r\left(\%\right)=\left(1-\frac{\sum^{\;}A_{D}}{\sum^{\;}A_{0}}\right)\;\times 100$$

$\sum^{\;}A_{D}$: sum of peak areas of EO component in the presence of dendrimers, $\sum^{\;}A_{0}$: sum of peak areas of EO component in free EO solution (control).

### 2.4. Screening of Six Encapsulation Parameters with Plackett–Burman Design

Plackett–Burman design was used to select the significant parameters for essential oils encapsulation. This design was applied to four combinations of dendrimers and EOs previously selected owing to their noticeable essential oil retention capacity (preliminary assays, data not shown but published soon). The combinations are: GD-PPI-3/CAN EO, GD-PAMAM-2/CIT EO, GAD-1/CAN EO, GCD-1/CIT EO. The independent parameters were set on the basis of those preliminary analyses, which considered the properties of the dendrimers for relative concentration and pH, the technical feasibility for rate of stirring, the solvent volume, and stirring time and the temperature which can be found in realistic agronomical conditions.

For each combination, a 12-run PBD was applied to evaluate six factors. Each variable was examined at two levels: –1 for the low level and +1 for the high level. Table 1 illustrates these parameters and the corresponding levels used. The values of two levels were set according to our previous preliminary experimental results. In Table 2, representing PBD and experimental results, data listed indicate the variations in retention rate between each combination of dendrimers-Eos, depending on the treatment. Negative values indicated that the opposite effect is observed: presence of dendrimers increase the volatility of EOs.

### 2.5. Optimization of Two Encapsulation Parameters by Response Surface Methodology

Based on the results of the PDB design, only the most influential parameters on the encapsulation reaction have been selected for further optimization through response surface methodology. Experiments were performed according to a design with two parameters and three levels for each parameter [25]. Two blocks have been used to cover the potential heterogeneity during the course of the experiment. The selected independent variables were stirring rate (R) and stirring duration (D). The experimental design in the actual levels is shown in Table 3. As for PBD, variations in retention rate between each couple dendrimers-EOs were recorded. In RSM experimental results (Table 4), negative percentage of retention notifies an increase in EOs volatility in presence of dendrimers.

Maximums were represented with contour plots.

### 2.6. Data Analysis

PBD and RSM were designed and processed using Minitab^®^ 19 software [25].

## 3. Results and Discussion

### 3.1. Volatiles Profiles and Major Components of EOs

Chromatograms obtained by DHS-GS-MS for encapsulation optimizations show the volatile profiles of both EOs in Figure 2 and Figure 3. Major compounds have been identified as it was previously mentioned [12]. On these figures, chromatograms of control and encapsulation solutions are overlaid which show that the only difference found is in the height (and peak area) of all compounds. Therefore, profiles were similar in the presence and absence of dendrimers. A thorough examination of the retention rate of each compound in Table 5 allows to observe that chemical structures and volumes of the major components of cinnamon EOs (volumes from 210 to 377 Å^3^) are more variable than in citronella EOs (volumes from 270 to 303 Å^3^), which seems to affect somewhat the profile (12% retention rate variations between eugenol and β-caryophyllene) 

### 3.2. Influence of Parameters with PBD

In the present study, the dendrimer/EOs complexes were successfully prepared by a simple solubilization and stirring in controlled conditions. To minimize the experimental runs and times for the screening of the encapsulation parameters, the PBD was applied on the basis of two coded levels of the six independent variables, resulting in twelve experiments (Table 2).

Analysis of PBD has been done for each couple dendrimer/EO (Table 6) which showed that almost no one had a variable influencing significantly the encapsulation rate (*p* < 0.05). However, the meta-analysis of all results and a particular attention at the ranking of variables show that time and rate of stirring appeared important in the encapsulation process. Considering that, it seems the lack of significance of these results reveals that the influence had been attenuated by the variability among repetitions in the manipulations. Both parameters (duration and rate of stirring) were selected for further optimization both with RSM.

### 3.3. Rate and Duration Stirring Optimization with Response Surface Methodology

#### 3.3.1. GD-PPI-3/CAN

For the first studied combination of dendrimer/EO, initially settled parameters were not optimal to find a maximum (Figure 4A) so new ones were defined in Table 7. Figure 5A shows that the model with those parameters was significant, with F-value equal to 10.34 and *p*-value < 0.001. Despite a slight rejection of the lack-of-fit test (*p* = 0.022) the applied model presented a good fitting to the encapsulation efficiency response (Figure 5B).

As the model is trustworthy, we can focus on the influence and optimization of factors. Linear and square of each parameter were significant (*p*-value < 0.05), so they were both influencing the encapsulation rate following the curves independently because their interaction (D*R) was not significant (*p*-value = 0.245). The regression equation describing these mathematical relationships is: (2)$$\left(r\right)\;=\;22.6\;+\;6.30\;D\;+\;4.12\;R\;-\;7.08\;D^{2}\;-\;5.49\;R^{2}\;+\;2.23\;D\times R$$

Contour plot present in Figure 4B illustrates the level of parameters that allowed to reach the maximum of retention (>20%) which can be found with a stirring time between 240 and 420 min at a rate between 1500 and 2000 rpm.

#### 3.3.2. GD-PAMAM-2/CIT

Second studied combination of dendrimer/EO showed that the model was significant with an F-value of 6.07 and *p*-value is 0.001 (Figure 6A). In addition, Figure 6B revealed a good correspondence between the linear regression model of RSM and the experimental data despite a slight rejection of the lack-of-lit test (*p*-value = 0.011). As for the first combination, linear and square of each parameter were significant but not their respective interaction. The regression equation describing these mathematical relationships is: (3)$$\left(r\right)\;=\;13.03\;-\;3.54\;D\;-\;6.43\;R\;-\;3.69\;D^{2}\;-\;3.45\;R^{2}+\;3.40\;D\times R$$

Contour plot present in Figure 7. illustrates that a stirring during between 10 and 60 min at a rate between 150 and 1000 rpm allowed to reach the maximum of retention (>15%).

#### 3.3.3. GAD-1/CAN

Third studied combination of dendrimer/EO showed that the model is significant with an F-value of 7.06 and *p*-value < 0.001 (Figure 8A) and the lack-of-lit is non-significant (*p*-value = 0.645). In addition, Figure 8B reveals a good correspondence between the linear regression model of RSM and the experimental data. Linear and square of only the duration of stirring are significant (*p*-value of R is 0.175 and R2 is 0.258) and influence the encapsulation rate following the curves. The regression equation describing these mathematical relationships is: (4)$$\left(r\right)\;=\;0.93\;-\;7.98\;D\;+\;1.92\;R\;-\;5.56\;D^{2}\;-\;1.66\;R^{2}\;-\;1.79\;D\times R$$

Contour plot present in Figure 9 illustrates the level of parameters that allow to reach the maximum of retention even if this one is very low (>5%). The best results can be found with a stirring time between 10 and 30 min at a rate between 1500 and 2000 rpm.

#### 3.3.4. GCD-2/CIT

The last studied combination of dendrimer/EO showed that the model is significant with an F-value of 4.17 and *p*-value = 0.005 (Figure 10A) however, lack-of-lit is rejected with a *p*-value equal to 0.003 so results have to be discussed. Nevertheless, Figure 10B reveals a good correspondence between the linear regression model of RSM and experimental data which confirms the global correctness of the model. Only the linear effect rate of stirring was significant (*p*-value = 0.240) and the square effect of both parameters were significant. The regression equation describing these mathematical relationships is: (5)$$\left(r\right)\;=\;8.62\;+\;3.55\;D\;+\;7.41\;R\;-\;11.65\;D^{2}-\;6.77\;R^{2}-\;2.04\;D\times R$$

Contour plot present in Figure 11 illustrates the level of parameters that allowed to reach the maximum of retention (>10%) which was found with a stirring during around 60 min at a rate of 1500 rpm.

## 4. Conclusions

For the first time, essential oils encapsulation by bio-sourced dendrimers was successfully carried out, and this reaction was optimized using PBD and RSM. The first part proved that only the rate and the time of stirring influenced the retention rate among the six factors analyzed. The second part optimizes both factors for each couple dendrimer/EO and resulted in very different results. This is quite understandable considering the apolar nature of the EOs’ constituents and the differences of structure between dendrimers. Indeed, we can see in Figure 1 that even if all dendrimers contain glycerol or glycerol derivatives in the intern structure or on the periphery of the dendrimer, and a polar surface, the properties of the cores are different. On one point, the core of GD-PAMAM-2 is more polar than the GD-PPI-3′s one; on another point, some have strong steric hindrance and important electronic charge (GCD-2) while others are less energy-intensive (GAD-1). Previous study about encapsulation by dendrimers showed that the hydrodynamic radius of GD-PPI and GD-PAMAM influenced the encapsulation and that one occurred at the core level of dendrimers rather than at its periphery. Metal complexes were successfully encapsulated in the fourth and fifth generation of GD-PPI (around 25% of encapsulation rate), but not in the third probably because this one had a smaller hydrodynamic radius (2.81 nm) [20]. Organic compounds as β-estradiol, atrazine, diclofenac salt, or diuron have been also encapsulated in GD-PPI-4 and GD-PAMAM-3 up to 95% [18]. As the trans-cinnamaldehyde (Table 5), one of the major compounds responsible of herbicidal activity, is a smaller molecule than the previous encapsulated ones, it seems obvious that smaller dendrimer generations give here the best results for its encapsulation. Furthermore, this α,β-unsaturated aldehyde presents an important electronic density as the previous organic compounds used. It must be pointed out that chromatographic profiles were similar, for EOs encapsulated in dendrimers or not (control) which suggests that all compounds of each EOs were encapsulated in the same way (Figure 2 and Figure 3). It can be concluded that first the size of molecules encapsulated in comparison with size of intern cavities of dendrimers, and secondly the amount of free electron in the EOs (aromatic circle and double bonds promotes electrostatic interactions) appear to be principal factors influencing the EOs encapsulation within dendrimers [28].

In the optimized conditions, the best encapsulation rates varied from 5 to 40% depending on the dendrimer-EO combination (Table 8). The combination of GD-PPI-3 with cinnamon EO leads to the most promising results with an r = 40% when the stirring is long (6 h) and strong (1735 rpm). As there is no other study on encapsulation of EOs within GDs yet, comparing these results with previous results is not possible. However a comparison with other encapsulations techniques can be done: for example, dendrimers have a better encapsulation rate than the powder optimized by Huynh T. V. et al. who obtained 18% as optimum EO concentration [29]. On the opposite, the rate of encapsulation is quite lower than encapsulation by coacervation in gelatin optimized by Sutaphanit P. and Chitprasert P. (66.5 to 98.4%) but the release from these capsules is almost impossible (stable for 18 months storage) [30]. In another field of application, optimized encapsulation of gallic acid in calcium alginate microbeads was of the same order (42.8%) [31].

In the context of the use of dendrimer-EOs formulations as biopesticide, it is essential to go further in the study of the encapsulation rate with a dynamic study of the release of EOs by the dendrimer. It is also worthwhile to determine the stability and biological effects of the new biosourced herbicide formulation. In addition, it would be relevant to study with a more fundamental point of view the encapsulation of the selected pure compounds from EO like trans-cinnamaldehyde within GD-PPI-3 to better understand the interactions between EO constituent and dendrimer particularly through NMR studies. This work is in progress.

This article shows for the first time that it is possible to effectively encapsulate essential oils in dendrimers. Given the numerous biocidal properties of essential oils, this technique opens the road to numerous applications in agronomy but also in other sectors where a slow release of essential oils is being researched, such as in pharmaceuticals or in the food industry with the design of innovative packaging.

## Figures and Tables

**Figure 1 foods-10-00207-f001:**
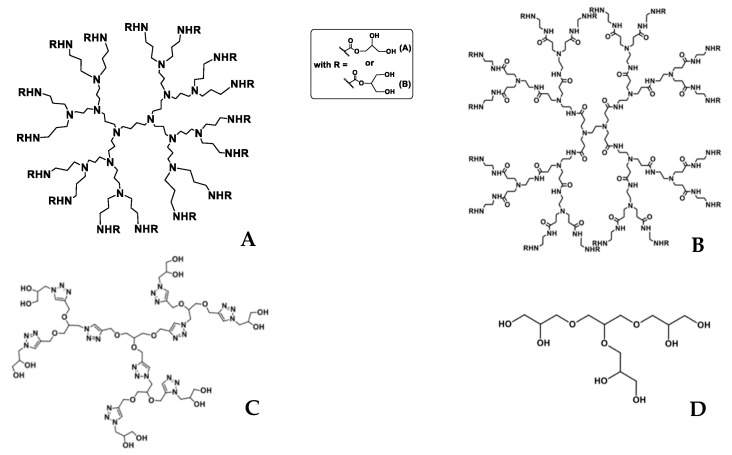
Structures of dendrimers: (**A**) Glycerodendrimers polypropylenimine 3rd generation (GD-PPI-3). (**B**) Glycerodendrimers polyamidoamine 2nd generation (GD-PAMAM). (**C**) Glyceroclikdendrimers 2nd generation (GCD-2). (**D**) Glyceroladendrimers 1st generation (GAD).

**Figure 2 foods-10-00207-f002:**
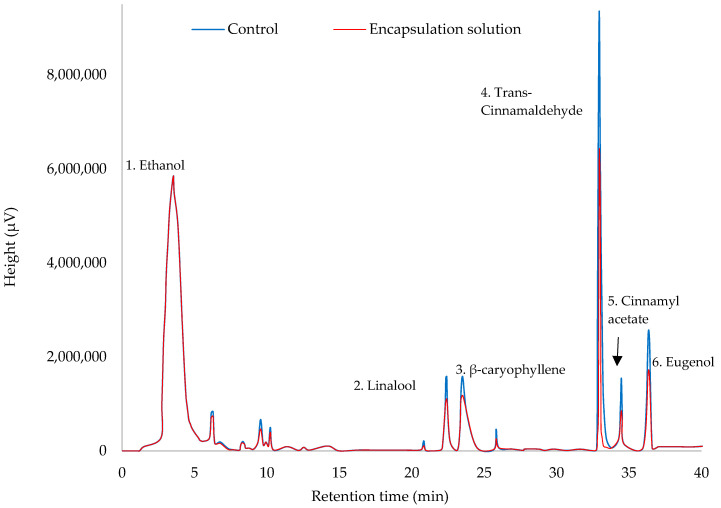
Overlaying of chromatographic analysis of free cinnamon EO (control) and cinnamon EO encapsulated within GD-PPI-3 under optimized conditions—(1) ethanol (sample solvent), (2) linalool, (3) β-caryophyllene, (4) trans-cinnamaldehyde, (5) cinnamyl acetate, (6) eugenol.

**Figure 3 foods-10-00207-f003:**
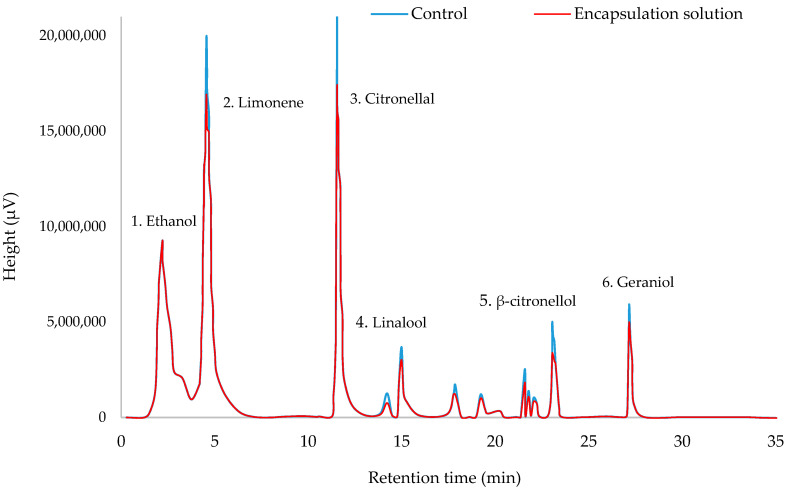
Overlaying of chromatographic analysis of free Java citronella EO (control) and Java citronella EO encapsulated within GD-PAMAM-2 under optimized conditions—(1) ethanol (sample solvent), (2) limonene, (3) citronellal, (4) linalool, (5) β-citronellol, (6) geraniol.

**Figure 4 foods-10-00207-f004:**
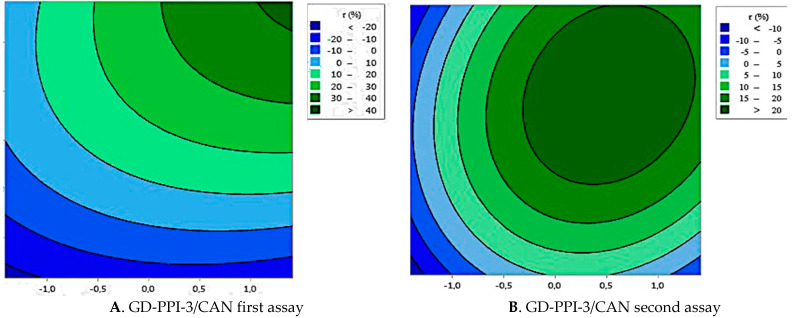
Contour plots showing the crossed effect of duration (D) and rate of stirring (R) on the retention rate (r) of cinnamon essential oil by GD-PPI-3 with the first sets of parameters (**A**) and the second one (**B**).

**Figure 5 foods-10-00207-f005:**
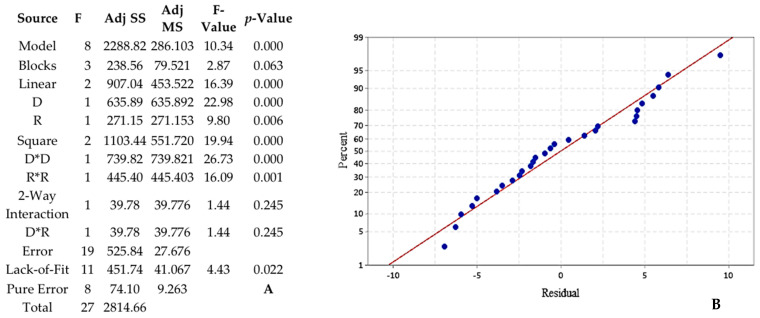
Analysis of variance (ANOVA) for the response surface methodology (RSM) (**A**) and normal probability plot of the residuals of GD-PPI−3/CAN EO (2) (**B**).

**Figure 6 foods-10-00207-f006:**
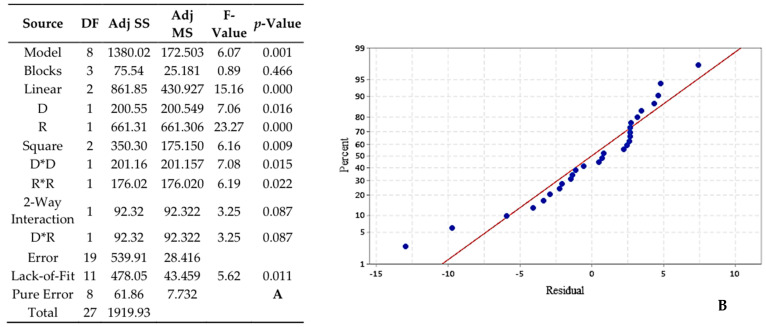
Analysis of variance (ANOVA) for the RSM (**A**) and normal probability plot of the residuals of GD-PAMAM-2/CIT EO (**B**).

**Figure 7 foods-10-00207-f007:**
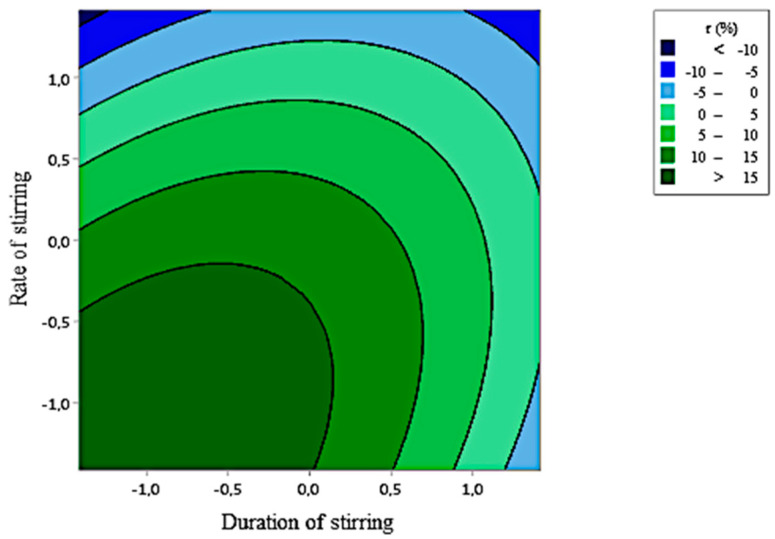
Contour plots showing the crossed effect of duration (D) and rate of stirring (R) on the retention rate (r) of citronella essential oil by GD-PAMAM-2.

**Figure 8 foods-10-00207-f008:**
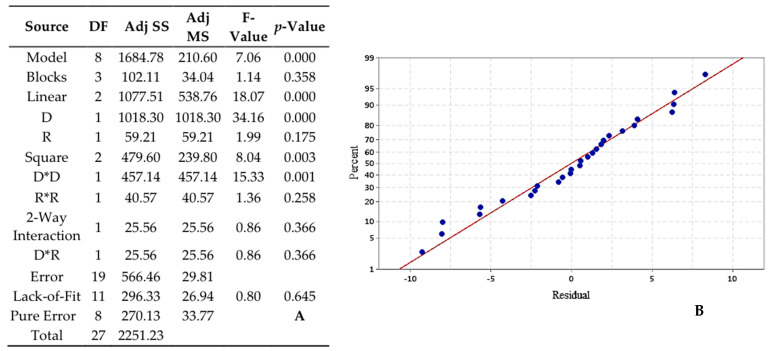
Analysis of variance (ANOVA) for the RSM (**A**) and normal probability plot of the residuals of GAD-1/CAN EO (**B**).

**Figure 9 foods-10-00207-f009:**
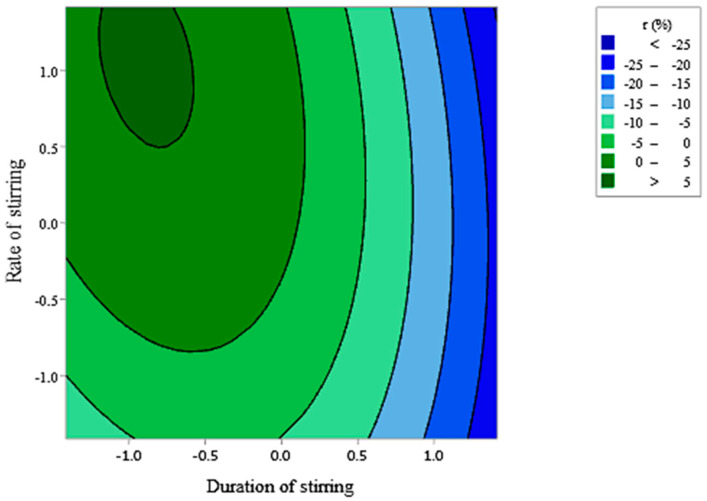
Contour plots showing the crossed effect of duration (D) and rate of stirring (R) on the retention rate (r) of cinnamon essential oil by GAD-1.

**Figure 10 foods-10-00207-f010:**
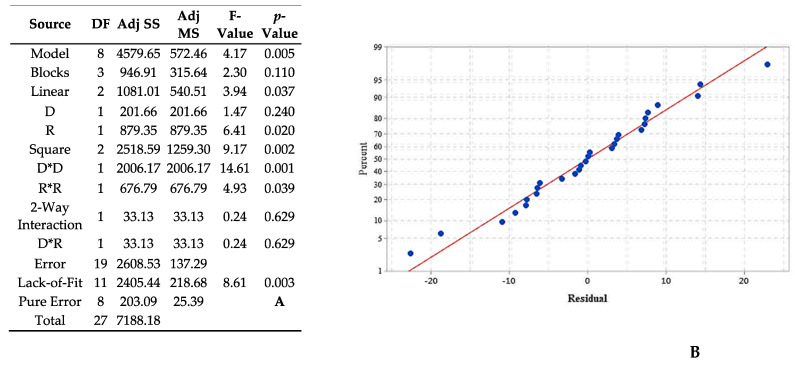
Analysis of variance (ANOVA) for the RSM (**A**) and normal probability plot of the residuals of GCD-2/CIT EO (**B**).

**Figure 11 foods-10-00207-f011:**
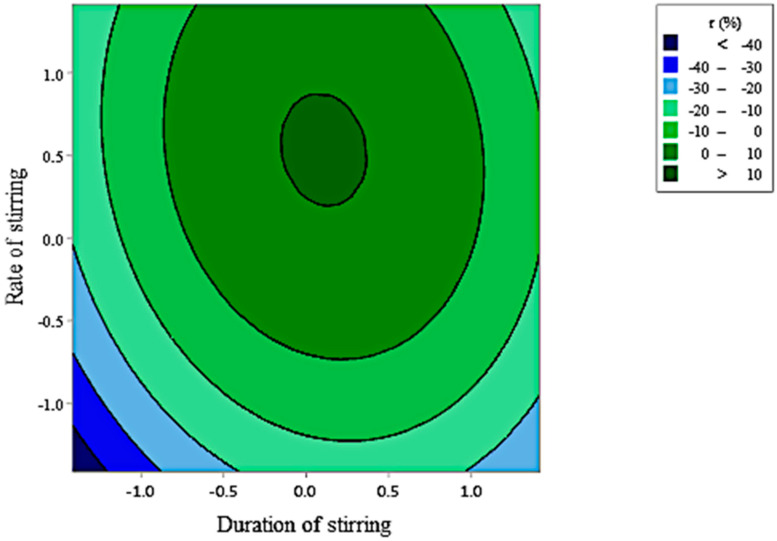
Contour plots showing the crossed effect of duration (D) and rate of stirring (R) on the retention rate (r) of citronella essential oil by GCD-2.

**Table 1 foods-10-00207-t001:** Factors and their levels selected for the Plackett–Burman design.

Factors	Symbol	Levels	
		−1	+1
Temperature (°C)	T	4	20
Rate of stirring (rpm)	R	150	800
Relative concentration (mg/mmol)	C	500	1500
Solvent volume (mL)	V	3	10
Stirring time (min)	D	10	240
pH	P	4	7

**Table 2 foods-10-00207-t002:** Experimental setting (12 runs) generated by Minitab^®^ 19 and retention rate for the fourth combinations of dendrimers and essential oils (Eos) (%, experimental).

Run	T	C	V	R	D	P	r (GD-PPI-3/CAN)	r (GD-PAMAM-2/CIT)	r (GAD-1/CAN)	r (GCD-2/CIT)
1	1	−1	1	−1	−1	−1	−30.79	2.29	32.04	54.11
2	1	1	−1	1	−1	−1	22.82	−95.35	18.69	−69.69
3	−1	1	1	−1	1	−1	−5.14	−23.76	−37.17	18.85
4	−1	1	1	1	−1	1	0.74	−10.22	−0.46	−9.39
5	−1	1	−1	−1	−1	1	9.22	3.44	−11.03	37.81
6	1	1	1	−1	1	1	−28.86	−29.84	21.52	−4.44
7	1	1	−1	1	1	−1	17.70	35.88	−40.48	−36.38
8	1	−1	−1	−1	1	1	−36.31	−26.39	−5.58	−19.96
9	1	−1	1	1	−1	1	27.33	−42.67	−3.88	22.40
10	−1	−1	1	1	1	−1	5.40	−104.65	−67.95	22.02
11	−1	−1	−1	−1	−1	−1	21.00	−29.09	30.83	59.72
12	−1	−1	−1	1	1	1	−24.11	−52.64	9.97	9.86

**Table 3 foods-10-00207-t003:** Factors and their levels selected for the Box–Behnken design (response surface methodology).

Factors	Symbol	Levels		
		−1	0	+1
Stirring time (min)	D	10	60	240
Rate of stirring (rpm)	R	150	1000	2000

**Table 4 foods-10-00207-t004:** Experimental setting (28 runs) generated by Minitab^®^ 19 and retention rate for the fourth combinations of dendrimers and EOs (%, experimental).

Run	D	R	r (GD-PPI-3/CAN)	r (GD-PAMAM-2/CIT)	r (GAD-1/CAN)	r (GCD-2/CIT)	r (GD-PPI-3/CAN (2))
**1**	1	1	−15.17	4.64	−22.80	−10.01	6.69
**2**	0	0	12.85	14.96	2.91	23.56	25.64
**3**	0	0	13.55	9.56	−1.67	15.67	19.47
**4**	−1	1	3.93	−7.55	7.03	8.09	5.57
**5**	1	1	39.55	0.78	−12.61	−0.64	18.32
**6**	0	0	16.69	11.21	6.94	7.49	26.96
**7**	−1	−1	−30.53	20.49	−4.51	−0.65	−4.92
**8**	0	−1.4	−23.54	6.90	4.43	−11.46	6.19
**9**	−1.4	0	−6.82	8.72	−0.08	4.07	3.41
**10**	1.4	0	12.00	4.71	−19.16	−19.77	22.07
**11**	0	1.4	21.45	3.10	3.42	0.24	15.74
**12**	0	0	3.56	18.83	5.36	8.54	29.07
**13**	0	0	8.25	12.36	−1.60	9.18	32.35
**14**	0	0	13.59	17.92	7.21	1.67	28.94
**15**	1	1	34.07	−15.68	−13.38	−0.17	16.49
**16**	0	0	19.15	11.85	−2.39	8.04	17.92
**17**	−1	−1	1.79	24.67	−3.37	−53.77	4.54
**18**	0	0	21.78	13.71	4.26	7.78	23.28
**19**	1	−1	6.78	5.63	−14.68	−21.57	4.76
**20**	0	0	19.84	8.08	−9.13	4.06	15.35
**21**	−1	1	15.20	0.36	10.89	−15.47	−0.43
**22**	0	0	21.18	13.71	−1.12	1.40	15.45
**23**	0	1.4	36.87	3.64	−9.70	8.28	26.11
**24**	1.4	0	34.60	5.06	−19.58	4.00	21.34
**25**	−1.4	0	15.16	10.99	−0.38	−39.21	−1.98
**26**	0	−1.4	1.84	17.76	−6.11	−8.88	9.50
**27**	0	0	29.50	13.46	−6.81	8.14	14.77
**28**	0	0	28.67	10.72	7.25	7.90	16.70

**Table 5 foods-10-00207-t005:** Chemical structures and calculated molecular volumes of the major compounds of cinnamon and Java citronella EOs; and their individual retention rate in the optimized encapsulation within dendrimers.

**Cinnamon EO**	**Linalool** 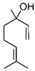	**β** **-Caryophyllene** 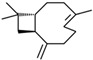	**Trans-Cinnamaldehyde** 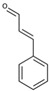	**Cinnamyl Acetate** 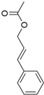	**Eugenol** 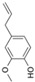	**Mean All Components**
Volume (Å^3^) *	294	377	210	279	257	
r (GAD-1)	9.18%	8.76%	10.93%	11.73%	12.21%	10.89%
r (GD-PPI-3)	28.21%	26.59%	35.99%	32.82%	38.97%	32.35%
**Citronella EO**	**Limonene** 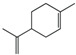	**Citronellal** 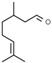	**Linalool** 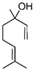	**β-citronellol** 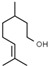	**Geraniol** 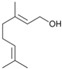	**Mean All Components**
Volume (Å^3^) *	270	297	294	303	291	
r (GCD-2)	11.76%	13.22%	12.98%	14.45%	13.65%	13.56%
r (GD-PAMAM-2)	22.99%	23.67%	21.92%	24.01%	23.51%	24.67%

* V = M/dN_A_ with M: molecular weight; d: density; N_A_: Avogadro’s number [27].

**Table 6 foods-10-00207-t006:** Analyses of variance (ANOVA) of Plackett–Burman screening design experiments.

GD-PPI-3/CAN					
	Effect Size	Coefficient	Std Error	F-Value	*p*-Value
Constant		−2.86	7.28	0.74	0.643
T	3.64	−1.82	7.28	0.74	0.643
C	11.22	5.61	7.28	0.06	0.813
V	−4.71	−2.36	7.28	0.59	0.476
R	17.9	8.95	7.28	0.1	0.759
D	−13.36	−6.68	7.85	1.51	0.274
P	−16.06	−8.03	7.28	0.72	0.434
**GD-PAMAM−2/CIT**					
Constant		−31.6	14.9	0.3	0.913
T	11.1	5.6	14.9	0.14	0.723
C	23.2	11.6	14.9	0.61	0.47
V	−6.5	−3.2	14.9	0.05	0.837
R	−28.7	−14.4	14.9	0.93	0.379
D	−6	−3	16	0.03	0.859
P	8.4	4.2	14.9	0.08	0.789
**GAD-1/CAN**					
Constant		−7.34	6.11	3.17	0.113
T	22.11	11.06	6.11	3.27	0.113
C	2.04	1.02	6.11	0.03	0.13
V	−3.96	−1.98	6.11	0.1	0.874
R	−28.56	−14.28	6.11	5.46	0.759
D	−45.6	−22.8	6.59	11.98	0.067
P	6.67	3.33	6.11	0.3	0.018
**GCD-2/CIT**					
Constant		5.41	6.93	3.79	0.083
T	−28.8	−14.4	6.93	3.79	0.083
C	−31.9	−15.95	6.93	4.32	0.092
V	23.7	11.85	6.93	5.3	0.07
R	−37.88	−18.94	6.93	2.93	0.148
D	−20.03	−10.02	7.47	7.48	0.041
P	−5.4	−2.7	6.93	1.8	0.237

**Table 7 foods-10-00207-t007:** Factors and their levels selected for the second assay of Box–Behnken design (response surface methodology) for the GD-PPI-3/CAN EO encapsulation.

Factors	Symbol	Levels		
		−1	0	+1
Stirring time (min)	D	60	240	420
Rate of stirring (rpm)	R	100	1500	2000

**Table 8 foods-10-00207-t008:** Optimized values of stirring rate and time for all combinations obtained using RSMs.

Combination	Stirring Rate	Stirring Time	Encapsulation Rate
GD-PPI-3/CAN EO	1735 rpm	366 min	39.92%
GD-PAMAM-2/CIT EO	120 rpm	10 min	19.93%
GAD-1/CAN EO	2142 rpm	9 min	9.75%
GCD-2/CIT EO	1528 rpm	65 min	10.78%

## Data Availability

The data presented in this study are available on request from the corresponding author.

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
