# Peer review of "Use of New Glycerol-Based Dendrimers for Essential Oils Encapsulation: Optimization of Stirring Time and Rate Using a Plackett—Burman Design and a Surface Response Methodology"

_foods, 2021, doi:10.3390/foods10020207_

Round 1
Reviewer 1 Report
The paper looks interesting by means of scientific topics. Unfortunately, in my opinion the paper should not be published in Foods. It is not because of lack of scientific soundness but I think the profile of the journal does not fit to the paper content. I advise to publish the paper in agricultural journal (e.g. Agriculture from MDPI).
Additionally going into the paper details I adore the optimization done by Authors but I think final equation as a result of optimization should be shown. It will also make the optimal parameter choice much easier. The second thing is the choice of EO. As Authors said oils are mixtures different active compounds. According to that I think some model study for predominant compound (or the most essential one) should be done first.
I also wonder why Authors did not perform any experiments on EO release from microcapsule in real “field experiment”. Modeling is a one thing but without real experiments seems to be useless.
My comments on the paper comes from a similar experiments done in food technology but for sure agricultural conditions vary so the paper should be reviewed by agriculture specialist before publication.
Reviewer 2 Report
The authors study nanoencapsulation of Cinamon and Java citronelle EOs in glycerol dendrites, suggesting that such nanocapsules may be used as natural herbicides.
This is a quite interesting work that needs to be improved by including more information in different sections:
- little information is provided in the introduction regarding the trully effect of such EOS as herbicides, more information is needed: previous research, uses, dose to be applied, needed contact time. Most herbicides show their effects 24 h after application. Do we really need to encapsulate them? What is the previous experience with both tested EOs. If they disappear too quickly which is the minimum volatilization time to be achieved wiht the nanocapsules?
- in materials and methods section very little information is provided on the procedures, please try to add some more information, even basic information on nanoencapsulation procedures
- volatile profile of EOs and nanocapsules are evaluated to determine the % of EO. The procedure is not clear to me, peak wihtout dendrimers are refered to control EOs? if so indicate as such instead of 'sum of peak areas without dendrimers'. Non released volatiles are those considered as encapsulated.
- there are no reported volatile profiles, please provide the volatile profile of the used EOs, as well as those of the optimized experiment. It would be really interesting to see if all volatiles are equally encapsulated or some compounds are more succesfully encapsulated than other. Please provide the tables and commento on these observations. (so you would be able to support your comments on the 'size of molecules' theory.
Round 2
Reviewer 1 Report
I am still not fully convinced about finding the article within the scope of the journal, however the authors have improved the article to be more food related. Additionally, a scientific improvement was made as suggested by the reviewer. So I can designate the article as Foods-ready
Reviewer 2 Report
The authors have properly addressed all raised questions, in my opinion this interesting mansucript is now much easier to unserstand